# A Fault Diagnosis Model for Coaxial-Rotor Unit Using Bidirectional Gate Recurrent Unit and Highway Network

Zhaoqin Peng [1], Kunyu Dong [1], Yan Wang [2,*] and Xucong Huang [1]

[1]  School of Automation Science and Electrical Engineering, Beihang University, Beijing 100191, China; pengzhaoqin@buaa.edu.cn (Z.P.); dongkunyu@buaa.edu.cn (K.D.); huangxucong@buaa.edu.cn (X.H.)
[2]  School of Transportation Science and Engineering, Beihang University, Beijing 100191, China
*  Correspondence: wybuaa@buaa.edu.cn

**Abstract:** A turbojet engine is the most significant part of an Internal Combustion Engine (ICE) for Hybrid Electric Vehicles. Specifically, the coaxial-rotor unit is the key component, whose performance largely affects the working efficiency. Thereby, the fault diagnosis methods for coaxial-rotor units is a main focus. In line with our test results, the bearing circlip is the most vulnerable element while rotating. Moreover, the low-speed rotating fault diagnosis is even challenging for current methods. Since the fault diagnosis on the bearing circlip of coaxial-rotor units is absent, this paper establishes a test rig on a running coaxial-rotor unit under different working conditions. The three-directional vibration signals are collected and analyzed to demonstrate the working states. On the task of bearing circlip failure classification, a deep-learning-based model using the Bidirectional Gate Recurrent Unit and the Highway Network is developed, which is capable of capturing hidden features and removing unrelated information. For working performance evaluation, experiments on the data of different rotating speeds are carried out. Among all the fault diagnosis methods, our model is the best approach and achieves an average accuracy of 99.4%. The encouraging results reveal that the proposed model is effective in both the high-speed and low-speed fault diagnosis of bearing circlip malfunction.

**Keywords:** turbojet engine; coaxial-rotor unit; fault diagnosis; Bidirectional Gate Recurrent Unit; Highway Network



## 1. Introduction

The automobile industry faces the challenge of saving energy and eliminating environmental damage, whilst delivering more goods and passengers to achieve a longer driving range [1]. In line with the ongoing advancements, electric vehicles, whose primitive form is proposed in 1900 s, have caused a revival of interest in vehicle engineering [2]. Based on the developing potential of electric vehicles and batteries, pure electric vehicles will ultimately meet the demands of high energy efficiency and low-pollutant emission. Despite its high energy efficiency and low-pollutant emission, the pure electric vehicle is currently a secondary choice for long-range driving restricted to battery storage capacity [3]. That is, these vehicle types are not likely to be widely used within the short term. Encouragingly, Hybrid Electric Vehicles (HEVs) bridge the gap between conventional vehicles and pure electric vehicles, which are viable means of transportation at present [4]. An HEV has an Internal Combustion Engine (ICE) as the main power source and an electric drive (i.e., a battery and an electric motor) to improve fuel economy [5]. Specifically, the battery required to power the electric motor is charged by the ICE without any external source. As such, almost all auto makers provide a hybrid version of their popular models such as Audi A8 Hybrid, Ford Fusion, Honda Civic Hybrid, and Hyundai Sonata Hybrid [6].

Typically, a turbojet engine, integrating with an electric generator, is employed in an ICE to supply stable and reliable power [7]. As long as the ICE aims to supply the

steady-state power requirements, the turbojet engine, integrated with the electric generator, is employed, which is deemed best able to meet the demands of high power, light weight, and simple structure. Under the working condition of high temperature, high pressure, and high vibration, each component of the turbojet engine has to be sufficiently robust to failures [8]. The coaxial-rotor unit is one such key component, and it improves the efficiency of energy conversion to a large extent [9]. Effort on coaxial-rotor unit is ongoing to not just enhance its reliability but also investigate its working states. Thereby, fault diagnosis, which aims to detect and identify any kinds of potential abnormalities and faults of the working turbojet engine, is highlighted [10,11].

Notwithstanding, the high-speed rotary machinery often runs under time-varying speed, which results in a huge challenge for fault diagnosis [12]. Theoretically, fault diagnosis aims to analyze the consistency of the obtained underlying knowledge and the system characteristics measured from the monitored data [13,14]. In most cases, high-speed rotary machinery often runs under time-varying speed, which results in a huge challenge for fault diagnosis. Because the establishment of working conditions is demanding, few experiments are carried out on real laboratory coaxial-rotor units. For instance, the high rotating speed fluctuation of mechanical equipment may lead to an unpredictable frequency and the amplitude modulation of the signal [15]. Previous works have been engaged primarily in system modeling and simulating. In such work, the working state differs from the actual mode because it focuses on the target working parameters instead of reconstructing the working scenarios. Thus, fault diagnosis approaches, which are formulated dedicatedly for simulation models, cannot be directly applied to practical fault identification [7,16].

According to the high-speed rotating demand of the coaxial-rotor unit for the turbojet engine, issues of fault diagnosis are far from solved. Therefore, the objective of this paper is to detect and identify the fault of the rotating unit. Testing on an actual coaxial-rotor unit is carried out to reproduce the actual working states of the turbojet engine and generate fault data. A set of sensing devices is employed for working condition monitoring. The Hilbert–Huang Transform (HHT) is implemented, which aims to investigate the fault mechanism. A deep-learning-based framework is thus proposed for fault diagnosis. The Gate Recurrent Unit (GRU) [17,18] with the Highway Network [19,20] is applied to realize the motivation in a novel way. Our method is capable of exploiting the basic characteristics to represent the failures via the sensing signals. The contributions of this paper are threefold and can be summarized as follows:

(1) A test rig for coaxial-rotor unit fault generation is established, based on which the working properties under different conditions are analyzed. Specifically, the bearing circlip failure is pronounced.
(2) A novel fault diagnosis method is established, integrating the Bidirectional Gated Recurrent Unit and Highway Network. Our model aims to not just capture the internal features of the working state but also identify the faults of the coaxial-rotor unit with various rotating speeds.
(3) In line with the practical working conditions, experiments are carried out on coaxial-rotor unit vibration signals to verify the effectiveness of the proposed model. Our model achieves decent working performance in the test.

## 2. Problem Formulation

### 2.1. Structure of Coaxial-Rotor Unit

Figure 1 presents the schematic of a turbojet engine. The major components of the engine are illustrated. The air from upstream comes through the centrifugal impeller of the compressor and reaches the diffuser. Via both the axial and radial diffusing, the air enters into the combustion chamber by passing the airhole. The combustion process is triggered by the ignition of the igniter, with the air mixing with the fuel. Hereafter, the high-pressure and high-temperature gas expands towards the turbine, which provides power to drive the spindle, the turbine, and the compressor rotating at high speed. Subsequently, the gas goes

into the jet nozzle at lower pressure and a higher speed. In this way, the thrust is produced whilst the exhaust is discharged to the atmosphere [21].

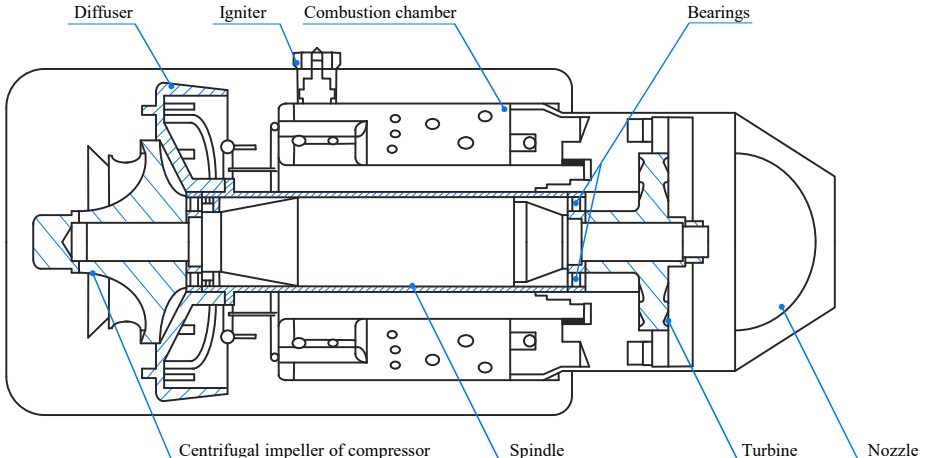

**Figure 1.** A schematic diagram of a turbojet engine.

We shall thus detach the structure of turbojet engine to resolve the coaxial-rotor unit. According to Figure 2, regardless of the outer case, the coaxial-rotor unit consists of a spindle, the high-speed rotors, and the bearings. Apparently, in this system, the rotors are mainly the turbine and the compressor. Meanwhile, there is a ball bearing on each end of the rotor. Generally, the bearing is made up of high-speed ceramic with the inner and outer ring of bearing steel. An example of a coaxial-rotor unit is exhibited in Figure 3. The rotating speed of the coaxial-rotor unit is over 10,000 r/min, while the highest speed reaching 100,000 r/min. Due to the working principle of the turbojet engine, the bearings are set to protect the rotors and, in turn, are most likely to break down during working. Based on the previous investigations, the bearing failure is the most probable failure mode in practical use [22–24].

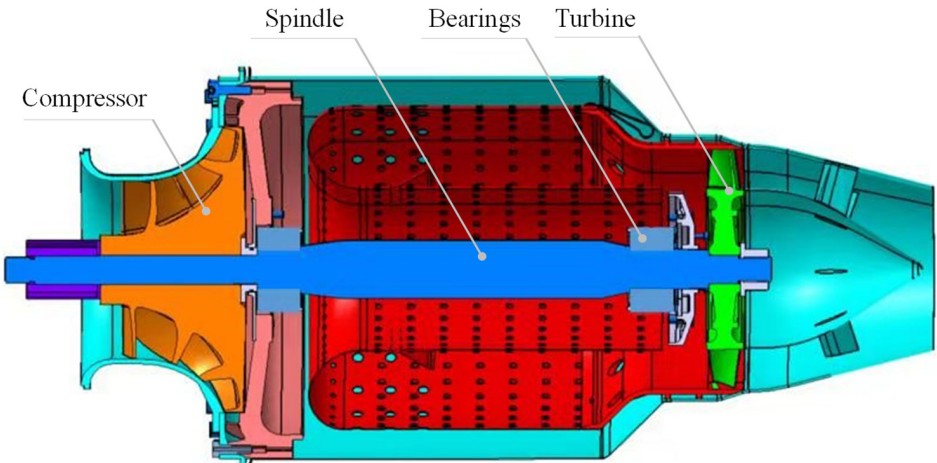

**Figure 2.** The structure of a coaxial-rotor unit.

### 2.2. System Establishment

As pointed out in the Introduction, the testing on the coaxial-rotor unit of the turbojet engine is a complex and interrelated process. For the task of fault diagnosis, the actual coaxial-rotor unit is built up and operated in various working states; see Figure 4. Note that we aim to concentrate on the condition of the coaxial-rotor unit, other components of the turbojet engine, including the combustion chamber and the ignition devices, are removed. The outlook of the coaxial-rotor unit in our experiment is presented in Figure 5a.

Considering the impact from the combustion, high-temperature gas is generated and insufflated to simulate the practical working environment. The gas generator is attached to the coaxial-rotor unit via the pipelines seen in Figure 5b. Notably, the rotating speed varies due to the control of the air–gas ratio inside the gas generator. That is, by adjusting the proportional valve of the flow meter, the inputs of gas and air are regulated via a controlling module (Figure 5d), and thus their mixing ratio is revised. In this way, the rotation of the coaxial-rotor can be determined. For instance, the rotating speed reaches 20,000 r/min at an air–gas ratio of 26.8:4.5. A lubricating oil condition unit and a cooling water condition unit are employed to facilitate the system operation. Besides, the temperature of the exhaust is set as 50 °C at the initial stage.

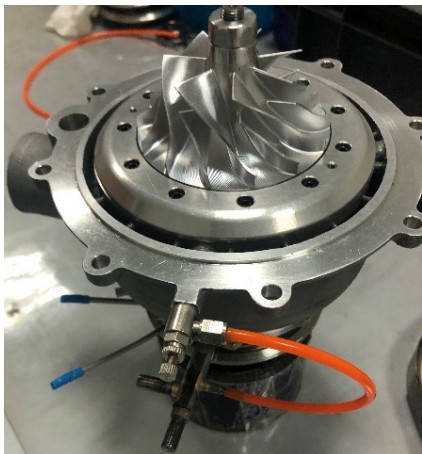

**Figure 3.** An example of a coaxial-rotor unit.

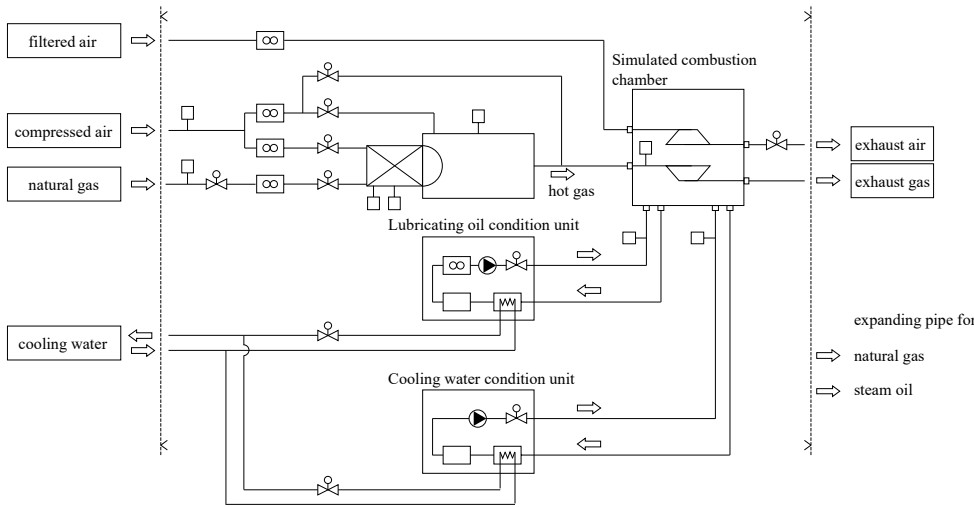

**Figure 4.** The schematic layout of the coaxial-rotor unit.

Sensing elements for detecting the working parameters are installed alongside the working components within the system. In the domain of measuring the rotating machinery, the vibration signal is most pronounced for characterizing the operational status directly. According to statistics, more than 70% of faults are manifested in the form of vibration [25]. In this way, to diagnose the potential faults, a three-axis vibrating acceleration sensor is attached to the rotor case. The specific installation position and sensor structure are shown in Figure 5c. The vibration of three directions—axial, radial, and tangential—is detected during working phases. With the power supplied, the vibrating signals are gathered by Integrated Circuits Piezoelectric (ICP) sensors 356A33 designed by PCB Piezotronics Inc. [26]. Sensing signals are transmitted through the 10-m low-noise cables. The data collecting and recording are implemented on an LMS830A analyzer. The measurement

frequency is as high as 50 k Hz for dynamic signal detection. The signal acquisition time is 120 s and involves a whole work phase of the coaxial-rotor unit. The three-directional vibration signals are displayed in real time and kept for further analysis. In addition, a tungsten–rhenium thermocouple temperature sensor measures the temperature of the hot gas whilst a Hall speed sensor monitors the rotating. Figure 5 shows the photograph of the major components of the system. The specifications of the corresponding measurement devices are given in Table 1.

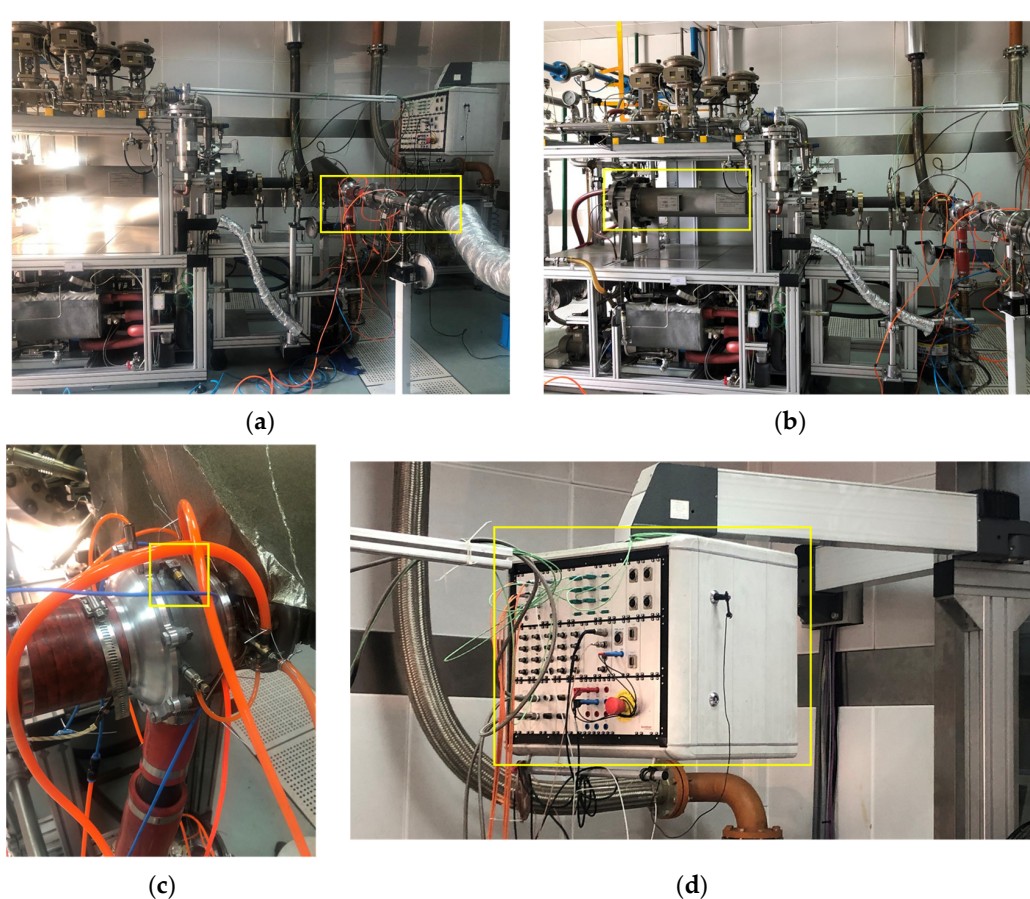

**Figure 5.** A photograph of the test rig. (**a**) The coaxial-rotor unit; (**b**) the gas generator; (**c**) the vibration sensor; and (**d**) the air–gas ratio controlling module.

**Table 1.** The specification of the measurement devices.

| Device | Model | Range | Precision | Quantity |
|---|---|---|---|---|
| Vibration sensor | ICP 356A33 | (−500~+500) g | 10 mg/V | 3 |
| Temperature sensor | W-Re5/26 | (0~1800) | 0.5% F.S. | 1 |
| Speed sensor | SS443A | (0–20 k) Hz | 1% F.S. | 1 |
| Vibration analyzer | LMS830A | (0–102.4 k) Hz | 1% F.S. | 1 |

We find that the circlip of the bearing is the most vulnerable part during working; see Figure 6. Notably, according to recent publications, a great deal of attention is paid to the fault diagnosis of bearings of rotating machinery [27]. Research on bearing circlip failures is currently absent. Hence, we mainly focus on the identification of the bearing circlip malfunction of the coaxial-rotor unit in this work.

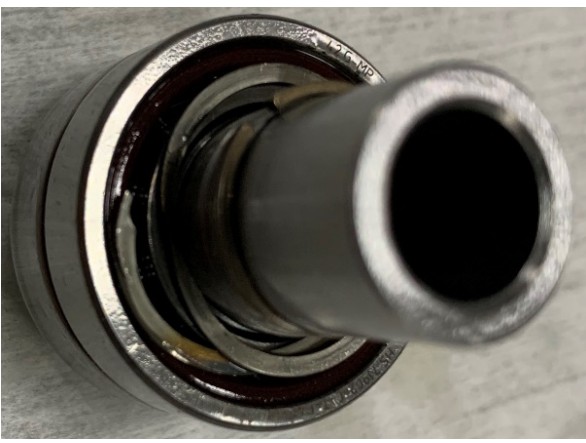

**Figure 6.** The failure of the coaxial-rotor bearing circlip.

## 3. Preliminary Analysis

Generally, the coaxial-rotor unit rotating speed of turbojet engine ranges from 20,000 r/min to 60,000 r/min, with an interval of 5000 r/min. The working-states-based varying-speed signals can be measured. As an example, the time-domain vibration of the three dimensions in the normal state and bearing circlip failure, with a speed of 40,000 r/min, is illustrated in Figures 7a and 7b, respectively.

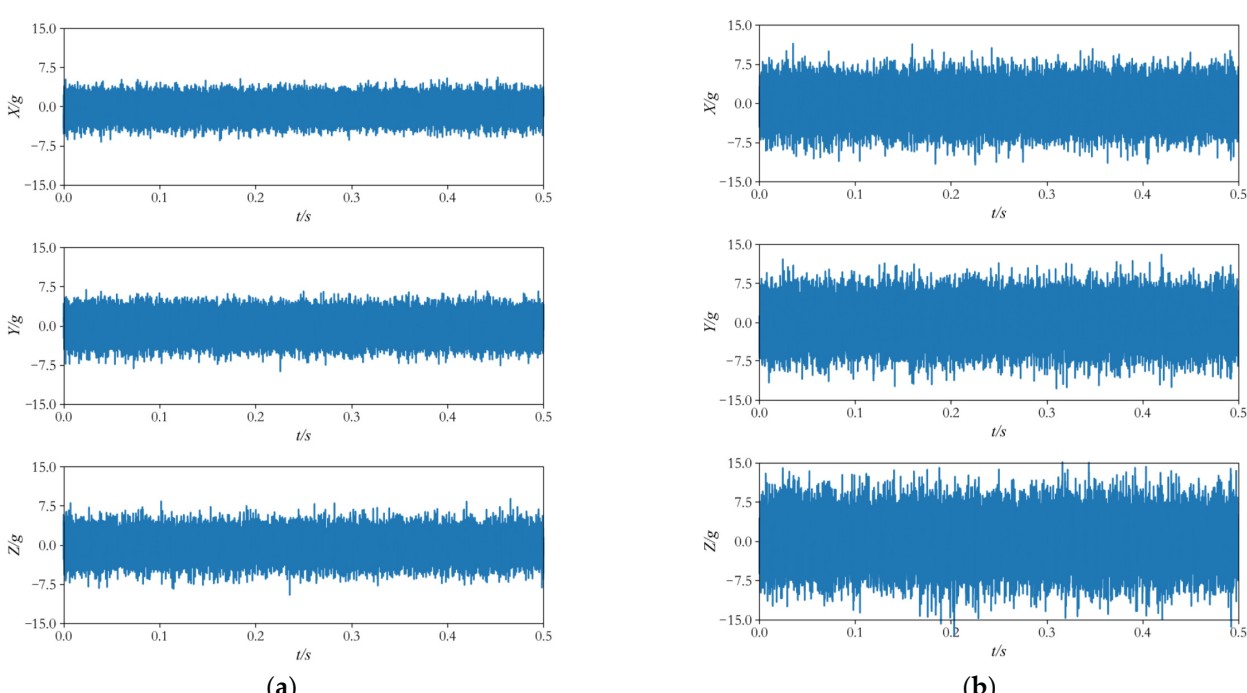

**Figure 7.** An illustration of the vibration signal. The X-Axial vibration, Y-Radial vibration, and Z-Tangential vibration. (**a**) The three-directional vibration in the normal state; (**b**) the three-directional vibration in bearing circlip fault.

Firstly, the vibrating data properties are investigated to describe the damage mechanism. For rotating machinery, the output power varies within the natural frequency band during the damage, which indicates the fault condition. At this stage, the Hilbert marginal spectrum is established to analyze the vibrating signals of various rotating speeds. To the best of our knowledge, the HHT is capable of tackling structure damage data in a variety of tasks [28,29].

Basically, the HHT consists of two main procedures, namely, Empirical Mode Decomposition (EMD) and Hilbert Spectrum Analysis (HSA). The EMD technique is taken to decompose a given signal into a number of Intrinsic Mode Functions (IMF) that provide much time-frequency information to the narrowband [30]. Each IMF component contains local features of different time scales and meets certain conditions, based on which fault features can be extracted [31]. Subsequently, the Hilbert transform is carried out on each IMF to obtain the corresponding Hilbert spectrum. That is, these IMFs are conveyed into the time-frequency domain. Then, the IMFs' entire Hilbert spectrum is aggregated to order the marginal spectrum derived in feature extraction [32]. In such a manner, the Hilbert marginal spectrum can be further used in fault diagnosis.

Let $x(t)$ be the vibration signals of the coaxial-rotor unit. The EMD is employed to decompose $x(t)$ to obtain a set of IMFs, which can be written as:

$$x(t) = \sum_{i=1}^{n} c_i(t) + r_n(t) \tag{1}$$

where $c_i(t)$ stands for the *ith* IMF; $n$ is the number of IMF components; and $r_n(t)$ is the residual function.

Then, the HHT is applied to each $c_i(t)$, so we have

$$H[c_i(t)] = \frac{1}{\pi} \int_{-\infty}^{\infty} \frac{c_i(\tau)}{t - \tau} d\tau \tag{2}$$

From Equation (2), $H[c_i(t)]$ and $c_i(t)$ can be integrated to obtain

$$z_i(t) = c_i(t) + j\, H[c_i(t)] = A_i(t) \cdot e^{j\varphi_i(t)} \tag{3}$$

where

$$A_i(t) = \sqrt{c_i{}^2(t) + H^2[c_i(t)]} \tag{4}$$

$$\varphi_i(t) = \arctan\left(\frac{H[c_i(t)]}{c_i(t)}\right) \tag{5}$$

and

$$w_i(t) = \frac{1}{2\pi} \frac{d\varphi_i(t)}{d(t)} \tag{6}$$

The parameters $A_i(t)$, $\varphi_i(t)$, and $w_i(t)$ represent the instantaneous amplitude, instantaneous phase, and instantaneous frequency of the *ith* IMF, respectively.

Considering that $r_n(t)$ is either a monotonic function or a constant, and no IMF component can be extracted from $r_n(t)$, the original signal can be rewritten as:

$$x(t) = Re \sum_{i=1}^{n} A_i(t) e^{j\varphi_i(t)} \tag{7}$$

The time-frequency-amplitude distribution is described using Hilbert spectrum, which is:

$$H(w,t) = Re \sum_{i=1}^{n} A_i(t) e^{j2\pi \int w_i(t) dt} \tag{8}$$

Further, the Hilbert marginal spectrum is obtained by the time integral of the Hilbert spectrum (Equation (9)), which delivers the amplitude distribution of each instantaneous frequency.

$$h(w) = \int_{-\infty}^{\infty} H(w,t) dt \tag{9}$$

Based on the Hilbert marginal spectrum, we define the Hilbert marginal spectrum energy as:

$$E(w) = h^2(w) \tag{10}$$

According to Equation (10), we see that the signal frequency characteristics is magnified in $E(w)$.

Since the vibration data of three directions are collected, the Hilbert marginal spectrum energy of the three-axis signal is the integration all these energies:

$$\hat{e} = \sum_{i=1}^{3} \partial_i x_i \tag{11}$$

where $\partial_i$ is the weighting coefficient of the *ith* Hilbert marginal spectrum energy and satisfies:

$$\sum_{i=1}^{3} \partial_i = 1 \tag{12}$$

The energy can be normalized to:

$$p_k = E_k / \sum_{k=1}^{m} E_k(\omega) \tag{13}$$

where $E_k$ represents the marginal spectrum energy value of the *kth* IMF component; $p_k$ is the energy normalization value; and $m$ is the order of the IMF.

Specifically, the Hilbert marginal spectrum energy of different rotating speeds is presented in Figure 8.

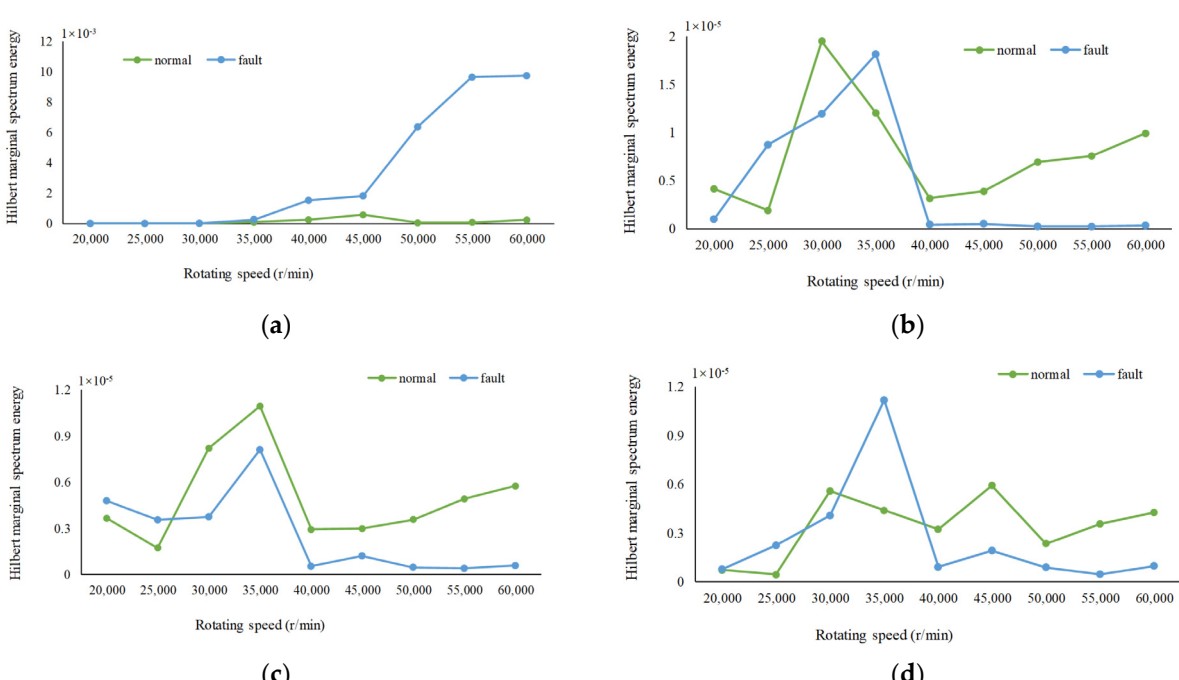

**Figure 8.** The Hilbert marginal spectrum energy distribution (**a**) IMF1; (**b**) IMF2; (**c**) IMF3; and (**d**) IMF4.

In Figure 8, one can easily see that the marginal spectral energy drops dramatically with the increase of IMF order. Notably, the marginal spectral energies of IMF2, IMF3, and IMF4 are negligible since their values are smaller than 10-5. For IMF1, the Hilbert marginal spectrum energies of the normal state and bearing circlip failure are distinguished when the rotating speed reaches 35,000 r/min. We thus infer that the fault identification at a low speed is challenging due to the similarity of marginal spectrum energy distribution.

## 4. Fault Diagnosis Approach

### 4.1. Model Architecture

Aiming to identify the working state of the coaxial-rotor unit under different conditions, a fault diagnosis approach based on a Bidirectional Gated Recurrent Unit (Bi-GRU) and the Highway Network is proposed. The architecture of our model is presented in Figure 9.

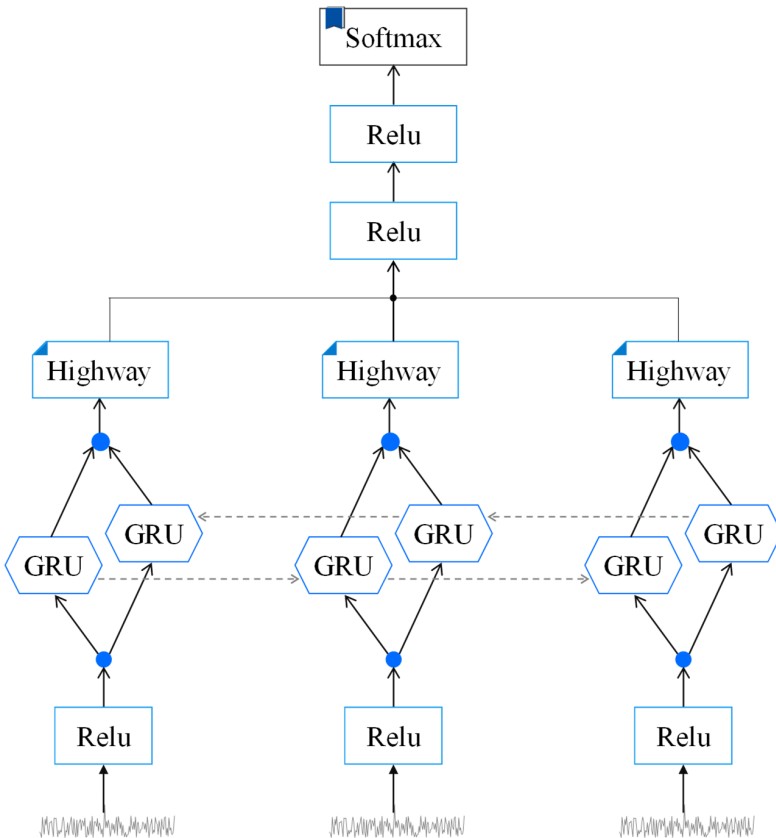

**Figure 9.** The mode architecture.

Let $x_i$, $y_i$, and $z_i$ represent the *ith* set of axial, radial, and tangential vibrating signals, respectively. $[x_i, y_i, z_i] \in R^n$ can be the inputs of the proposed fault diagnosis model. Each input vector is sent to a fully connected layer for feature extraction. Specifically, the active function Rectified Linear Unit (ReLU) is exploited to transform the input into feature-based sequence, which is expressed as:

$$x_i' = wx_i + b \tag{14}$$

$$X_i = ReLU(x_i') \tag{15}$$

where $w$ and $b$ are weight matrix and bias vector derived from model training.

Similarly, $Y_i$ and can be $Z_i$ obtained in the same manner. The features of three-direction vibration are thoroughly distilled.

Then, the Bi-GRUs are taken to extract the internal properties of the input data. We will now briefly describe the theory of GRU. For details, see, for example, [17,19]. GRU is a variation of the Recurrent Neural Network (RNN), which is widely applied to learn sequential characteristics from a sequential data stream [33]. Concretely, a GRU is established on the gating mechanism to modulate the existing memory and new memory: a reset gate determines the combination of new input and previous information, and an update gate selects previous information to preserve [34]. The sigmoid activation function computes both gate outputs, while tanh is used as the candidate hidden activation function. Distinctively, the GRU controls the information flow and exposes the full hidden

information without a memory unit [35]. In this way, the GRU is generally employed to obtain the hidden states of inputs.

In our model, a Bi-GRU related to sequence data is established. The input is fed into both a forward GRU and a backward GRU. These two GRUs of reverse timing are connected to one output. For the input $X_i$, we have that:

$$\begin{cases} \overrightarrow{h}_{X_i} = \overrightarrow{GRU}(X_i) \\ \overleftarrow{h}_{X_i} = \overleftarrow{GRU}(X_i) \\ h_{X_i} = \left[\overrightarrow{h}_{X_i}, \overleftarrow{h}_{X_i}\right] \end{cases} \tag{16}$$

where $h_{X_i}$ is the Bi-GRU output, consisting of forward hidden state $\overrightarrow{h}_{X_i}$ and backward hidden state $\overleftarrow{h}_{X_i}$. Likewise, for inputs $Y_i$ and $Z_i$, we can also compute $h_{Y_i}$ and $h_{Z_i}$.

Furthermore, the Highway Network is introduced to deal with the outputs of Bi-GRU (i.e., $h_{X_i}$, $h_{Y_i}$, and $h_{Z_i}$). In a Highway Network, a part of the inputs can be pass through the layer straightly, while others are revised using gating units [36].

Supposing that $T$ is the transform gate and $C$ is the carry gate, for an input vector $h_X$, the output of Highway Network is:

$$X_i^H = H(h_{X_i}, W_H) \cdot T(h_{X_i}, W_T) + h_{X_i} \cdot C(h_{X_i}, W_c) \tag{17}$$

The computation of $X_i^H$ is facilitated by setting $C = 1 - T$. Thereby, Equation (17) is modified to:

$$X_i^H = H(h_{X_i}, W_H) \cdot T(h_{X_i}, W_T) + h_{X_i} \cdot (1 - T) \cdot (h_{X_i}, W_c) \tag{18}$$

where $H$ is an affine transform followed by a non-linear activation function, and $W_H$, $W_T$, and $W_C$ are the parametric matrices optimized during training. Correspondingly, the outcomes $Y_i^H$ and $Z_i^H$ are computed as well.

With the application of Highway Network, more significant properties from vibrating signals are retained whilst irrelevant information properties are removed. The Highway Network outputs are concatenated and sent to the two-layer activation. The information propagation during this process is delivered as:

$$R_i = \left[X_i^H \oplus Y_i^H \oplus Z_i^H\right] \tag{19}$$

then

$$L_i = ReLU(R_i) \tag{20}$$

$$U_i = ReLU(L_i) \tag{21}$$

Lastly, fault identification is performed via a softmax classifier to obtain the probability distribution over the different working states, which is:

$$u_i = \widetilde{w_o} U_i + \widetilde{b_o} \tag{22}$$

$$o = softmax(u_i) = \frac{\exp(u_i)}{\sum_{k=1}^{c} \exp(u_i{}^k)} \tag{23}$$

where $\widetilde{w_o}$ is a trainable matrix, $\widetilde{b_o}$ is the offset vector, and $C$ is the number of working states.

### 4.2. Model Training

The training of our model is conducted by using the cross entropy and $L_2$ regularization as the loss function:

$$J = -\sum_{i=1}^{C} g_i log(p_i) + \lambda_r \left(\sum_{\theta \in \Theta} \theta^2\right) \tag{24}$$

where $g_i$ is the real working state distribution, and $p_i$ is the predicted one. The parameter $\lambda_r$ is the weight of $L_2$ regularization. The gradients and other parameters are updated through back propagation with the learning rate $\lambda_l$:

$$\Theta = \Theta - \lambda_l \frac{\partial J(\Theta)}{\partial \Theta} \tag{25}$$

## 5. Empirical Study

### 5.1. Experimental Setting

Fault classification experiments are conducted on data measured from coaxial-rotor units of real working conditions. The dataset contains three-directional vibration signals of both normal-state and bearing-circlip failure, with nine distinctive rotating speeds. The data are randomly split into training, validating, and testing according to the ratio 8:1:1.

Our model is finetuned using an ADAM optimizer with $\beta_1 = 0.9$, $\beta_2 = 0.999$, and $\varepsilon = 1 \times 10^{-8}$. All the parameter matrices are generated within the distribution of $U(-0.1, 0.1)$ randomly, while the bias is initialized as 0. The initialized parameters take 1000 times iteration for parameter optimization. Moreover, the hidden states dimension of the GRU cell is set as 256 with a learning rate of 0.001. The batch size and the training epoch are set to 10. The dropout rate is 0.5 to prevent overfitting. With respect to model training, the $L_2$-regularization weight is set as 0.0001.

### 5.2. Experimental Protocol

In this experiment, the confusion matrix is taken as the evaluating metric to verify the working performance of our model. In a confusion matrix (Table 2), the basic parameters, accuracy, precision, and recall can be computed [37,38]. On the one hand, the index accuracy denotes the correctlyclassified samples of all the samples, which is:

$$Accuracy = \frac{TP + TN}{TP + FP + FN + TN} \tag{26}$$

**Table 2.** The confusion matrix.

| Predicted/Actual | Actual: YES | Actual: NO |
|---|---|---|
| Predicted: YES | TP(True Positive) | FP(False Positive) |
| Predicted: NO | FN(False Negative) | TN(True Negative) |

On the other hand, precision and recall also provide a richer measure of classification performance. According to Equation (27), precision stands for the proportion of predicted positive cases that are correctly real positives:

$$Precision = \frac{TP}{TP + FP} \tag{27}$$

In contrast, recall refers to the proportion of real positives that are correctly predicted as positive:

$$Recall = \frac{TP}{TP + FN} \tag{28}$$

In this experiment, 'precision' can be taken as a key metric for working performance demonstration. Normal working condition indicates 'positive', and failure indicates 'negative'; it is more severe to classify a fault as a normal state than the opposite. Therefore, the accuracy is adopted as the primary evaluation index and precision as the auxiliary.

### 5.3. Main Results

The main purpose of the experiment is to verify the effectiveness of the proposed model for the aspect-level sentiment analysis. The following baseline methods are taken for comparison.

KNN (K—Nearest Neighbor): A simple machine learning algorithm for both linear and non-linear classification, which is insensitive to outliers.

SVM (Support Vector Machine): A supervised learning algorithm for generalized linear classification, which can prevent the issue of over fitting.

CNN (Convolutional Neural Network): A classical deep learning method, consisting of a convolutional layer, a pooling layer, and a fully connected layer for feature extraction.

LSTM (Long Short-Term Memory): A variant of RNN, which can detect the hidden states of the inputs.

GRU: As mentioned above, GRU is a variant of RNN that captures the hidden states using a gating mechanism and an activation function.

Bi-GRU: Bi-GRU is built based on a forward GRU and a backward GRU, to encode and decode the inputs, respectively.

Notably, five parameters are picked up as features and applied to KNN and SVM, which are the average value, standard deviation, maximum value, kurtosis, and IMF1 marginal spectral energy.

The fault diagnosis results of different rotation speeds are obtained. The test accuracy of all the working conditions is given in Table 3. The confusion matrix that presents test outcomes is shown in Figure 10. Notably, the values on the diagonal line refer to all the correctly recognized numbers of samples, corresponding to the accuracy of the test. The darker the color is, the higher the accuracy is.

In nine evaluation settings, our model obtains the best and most consistent performance in classification accuracy. As presented in Table 3, an average accuracy as high as 99.4% is obtained due to its distinctiveness in both high-speed and low-speed rotating fault diagnosis. Among all the approaches, SVM and KNN have relatively lower accuracy than the deep learning models because current machine learning models still have challenges in feature extracting and exploiting. A deep neural network is taken as a better approach than machine learning algorithms in capturing the working properties under different working conditions. Specifically, one can easily see that the accuracy of low-speed diagnosis is even lower, which validates the analysis results based on Hilbert marginal spectrum energy.

**Table 3.** The testing accuracy of the fault diagnosis.

| Rotating Speed (r/min) | KNN | SVM | CNN | LSTM | GRU | Bi-GRU | Our Model |
|---|---|---|---|---|---|---|---|
| 20,000 | 85.0% | 72.8% | 89.5% | 82.4% | 81.5% | 99.9% | **100%** |
| 25,000 | 75.0% | 64.4% | 86.7% | 99.3% | 94.0% | 95.9% | **99.4%** |
| 30,000 | 94.4% | 95.6% | 95.0% | 95.5% | 92.0% | 97.8% | **98.0%** |
| 35,000 | 81.1% | 72.2% | 100% | **100%** | 99.5% | 99.9% | 99.6% |
| 40,000 | 100% | 100% | 90.2% | **100%** | 98.3% | 99.9% | 99.2% |
| 45,000 | 100% | 100% | 94.5% | 100% | 100% | 100% | **100%** |
| 50,000 | 86.1% | 91.1% | 100% | 99.8% | 99.6% | 100% | **100%** |
| 55,000 | 77.8% | 76.7% | 99.4% | 98.5% | **100%** | 99.1% | 99.3% |
| 60,000 | 100% | 100% | 99.7% | **100%** | 99.7% | 99.9% | 99.1% |
| Average | 88.8% | 85.7% | 95.0% | 97.3% | 96.0% | 99.1% | **99.4%** |

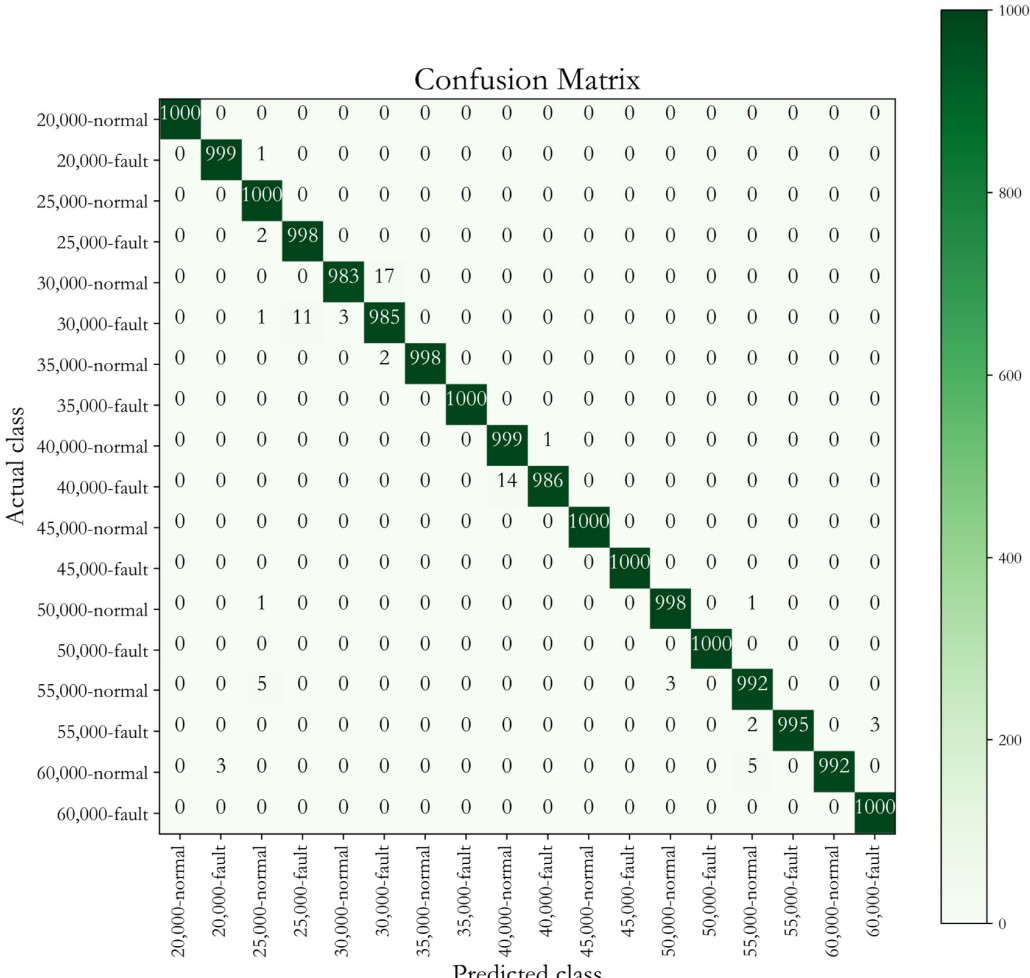

**Figure 10.** The confusion matrix for the fault classification.

### 6. Concluding Remarks

In this work, a deep-learning based fault diagnosis model is devised and deployed on the task of coaxial-rotor unit working state identification in a turbojet engine. To capture the essential characteristics of different states, the test rig is developed to reproduce the working condition and regenerate the fault data of the coaxial-rotor unit. Based on the test results, attention is paid to the faults on the bearing circlip, which are typically damaged at first. The three-direction vibration signals are measured and dedicatedly analyzed to characterize the working status. The HHT analysis is performed to study the failure mechanism of the bearing circlips. In line with the property of bearing circlip failure, a fault classification method, based on the integration of Bi-GRU and the Highway Network, is proposed. Basically, the GRU cells are capable of capturing the hidden information from inputs, while the Highway Network removes the unrelated features. The fault diagnosis performance of the proposed model is evaluated on nine different working conditions. An average accuracy of 99.4% of the proposed model is obtained, which outperforms the state-of-the-art methods. Specifically, our model also shows its superiority in low-speed fault classification. The experimental results indicate that our model obtains sufficient classification accuracy. That is, our model shows its distinctive superiority in dealing with the detected sensing signals precisely and thus can be applied to coaxial-rotor unit fault diagnosis tasks.

**Author Contributions:** Conceptualization, Z.P. and Y.W.; methodology, K.D.; software, K.D.; validation, K.D. and Z.P.; formal analysis, K.D. and X.H.; investigation, K.D. and X.H.; data curation, K.D.; writing—original draft preparation, Z.P.; writing—review and editing, Z.P.; visualization, K.D.; supervision, Z.P.; project administration, Y.W.; and funding acquisition, Y.W. All authors have read and agreed to the published version of the manuscript.

**Funding:** This research received no external funding.

**Institutional Review Board Statement:** Not applicable.

**Informed Consent Statement:** Not applicable.

**Data Availability Statement:** Not applicable.

**Conflicts of Interest:** The authors declare no conflict of interest.

**Abbreviations and Nomenclature**

| | |
|---|---|
| HEV | Hybrid Electric Vehicle |
| ICE | Internal Combustion Engine |
| GRU | Gate Recurrent Unit |
| ICP | Integrated Circuits Piezoelectric |
| PCB | Printed Circuit Board |
| HHT | Hilbert–Huang Transform |
| EMD | Empirical Mode Decomposition |
| HAS | Hilbert Spectrum Analysis |
| IMF | Intrinsic Mode Functions |
| Bi-GRU | Bidirectional Gated Recurrent Unit |
| ReLU | Rectified Linear Unit |
| RNN | Recurrent Neural Network |
| ADAM | Adaptive Moment Estimation |
| TP | True Positive |
| TN | True Negative |
| FP | False Positive |
| FN | False Negative |
| KNN | K-Nearest Neighbor |
| SVM | Support Vector Machine |
| CNN | Convolutional Neural Network |
| LSTM | Long Short-Term Memory |
| $c_i$ | i-th IMF |
| $r_n$ | n-th residual function |
| $A$ | instantaneous amplitude |
| $\varphi$ | instantaneous phase |
| $w$ | instantaneous frequency |
| $E$ | the marginal spectrum energy value of IMF |
| $p$ | energy normalization value |
| $h$ | output of Bi-GRU |
| $\overrightarrow{h}$ | forward hidden state |
| $\overleftarrow{h}$ | backward hidden state |
| $T$ | transform gate of Highway Network |
| $C$ | carry gate of Highway Network |
| $W$ | parametric matrices optimized during training |
| $\widetilde{w_o}$ | trainable matrix |
| $\tilde{b_o}$ | offset vector |
| $\lambda$ | weight of $L_2$ regularization |

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
