# Peer review of "A Fault Diagnosis Model for Coaxial-Rotor Unit Using Bidirectional Gate Recurrent Unit and Highway Network"

_machines, doi:10.3390/machines10050313_

Round 1
Reviewer 1 Report
A deep learning model for fault diagnosis has been proposed.
The advantages of the proposed method in comparison
with the other methods need to be highlighted.
Reviewer 2 Report
The paper presents an interesting study, with a right methodology and the manuscript is clear, well organized and structured, with experimental results, and the authors have worked exhaustively, taking care of the technical details.
It is opinion of the reviewer that the manuscript is very well but some suggestions could improve the paper:
- The introduction section should be improved, taking into account the more relevant studies on the subject to build a complete scientific framework.
- It is advisable to make an acronyms list.
- The manuscript format should be reviewed. For example, the text in Table 1 is not the same size.
- In the experimental section, the characteristics of the measured vibration signals should be included: Sampling frequency, measured time, number of measured vibration signals, ...
- The conclusions section can be extended and improved.
For these reasons, the reviewer suggests the manuscript for the publication after minor revisions.
Reviewer 3 Report
1) Throughout the paper text (as well as in the Abstract and Conclusion), the Authors should avoid the usage of the word “we” and the text should be written in neutral form. For example-in the Abstract-instead of: “we develop a deep-learning based model” should be used: “a deep-learning based model is developed”. The same should be implemented and corrected everywhere where occur in the paper.
2) The Abstract is too general and only descriptive. In the Abstract the Authors should add some of the most important results obtained in this research (its exact values), which cannot be found in other literature. Such addition will highlight the novelty of the presented paper already in the Abstract. Therefore, the Abstract requires re-arrangement and addition of the most important obtained results.
3) Figure 5 title – the last part of this figure is (d), not (b) as presented in the Figure title.
4) Section 2 – in this section should be added a complete list and at least a general specifications (accuracy and precision) of each used measuring device. As the measuring devices are essential for obtaining accurate and precise measurements, this data are very important. Some of the measuring devices are mentioned in this part, but mentioned data are required for all of them – these data can be also placed in the paper Appendix – the Authors can choose where is better to present them.
5) Section 2 – measurement setup is properly presented. However, there are missing data related to the measurement procedure and details about measuring process – which should be added in this Section.
6) Figure 8 – each figure part (IMF1 to IMF4) should be explained in a figure title. Also, for IMF3 and IMF4, the x-axis title (Rotating speed (r/min)) should be placed in the figure (as it is placed for IMF1 and IMF2).
7) Figure 10 is actually a Table – therefore, a correction is required. Also, all calls on this Table in the paper text should be corrected.
8) Line 317 – this is Figure 11, not Figure 13 (Figure 13 did not exist in the paper). A correction is required.
9) In the paper should be added a Nomenclature inside which will be listed and explained all abbreviations, symbols and markings used throughout the paper text. The Nomenclature will improve paper readability.
10) Figure 11 - Confusion matrix – should be explained more precise and more detail than is explained at the moment.
11) As the Abstract, the Conclusions section should also be improved with the most important obtained results (its exact values). Also the Conclusions seem to be too descriptive and general, without any details obtained in the presented analysis.
Final remarks: This is very interesting and innovative paper. After performing above mentioned corrections/additions, the paper should be accepted for publication.
